# Maternal Copy Number Imbalances in Non-Invasive Prenatal Testing: Do They Matter?

**DOI:** 10.3390/diagnostics12123056

**Published:** 2022-12-06

**Authors:** Michaela Hyblova, Andrej Gnip, Marcel Kucharik, Jaroslav Budis, Martina Sekelska, Gabriel Minarik

**Affiliations:** 1Medirex Group Academy n.o., Novozamocka 67, 949 05 Nitra, Slovakia; 2Trisomy Test s.r.o., Novozamocka 67, 949 05 Nitra, Slovakia; 3Medirex a.s., Galvaniho 17/C, 820 16 Bratislava, Slovakia; 4Geneton s.r.o., Ilkovicova 8, 841 04 Bratislava, Slovakia

**Keywords:** non-invasive prenatal testing (NIPT), genetic variation, copy number variations (CNVs), population databases, low-coverage whole-genome sequencing

## Abstract

Non-invasive prenatal testing (NIPT) has become a routine practice in screening for common aneuploidies of chromosomes 21, 18, and 13 and gonosomes X and Y in fetuses worldwide since 2015 and has even expanded to include smaller subchromosomal events. In fact, the fetal fraction represents only a small proportion of cell-free DNA on a predominant background of maternal DNA. Unlike fetal findings that have to be confirmed using invasive testing, it has been well documented that NIPT provides information on maternal mosaicism, occult malignancies, and hidden health conditions due to copy number variations (CNVs) with diagnostic resolution. Although large duplications or deletions associated with certain medical conditions or syndromes are usually well recognized and easy to interpret, very little is known about small, relatively common copy number variations on the order of a few hundred kilobases and their potential impact on human health. We analyzed data from 6422 NIPT patient samples with a CNV detection resolution of 200 kb for the maternal genome and identified 942 distinct CNVs; 328 occurred repeatedly. We defined them as multiple occurring variants (MOVs). We scrutinized the most common ones, compared them with frequencies in the gnomAD SVs v2.1, dbVar, and DGV population databases, and analyzed them with an emphasis on genomic content and potential association with specific phenotypes.

## 1. Introduction

Since 2013, NIPT has advanced from investigating common trisomies to reaching the subchromosomal or even monogenic levels for certain medical conditions [1,2]. The reporting of CNV imbalances in terms of duplications and deletions as incidental findings made through NIPT varies widely among different NIPT providers. Most providers disclose CNVs that are expected to be clinically relevant and potentially actionable. The positive predictive value (PPV) for CNVs is significantly lower than that of common trisomies [3,4], but it is still higher than that of sex chromosomal aneuploidies, particularly monosomy X [5]. Since the vast majority of cell-free (cf) DNA is derived from the maternal genome, genome-wide approaches primarily reveal maternal imbalances with much higher resolution than fetal imbalances, even with low-coverage genomic sequencing. Although NIPT was initially developed to detect medical conditions of the fetus, it eventually came to represent a liquid biopsy of the mother on a diagnostic level. Although most maternal CNVs can be considered common and benign according to ACMG criteria or in silico CNV prediction tools, they can also provide insight into the mechanisms of multifactorial diseases and potentially have clinical relevance. Globally collected NIPT data could be used for genome-wide association studies (GWAS) in large-scale epidemiological studies. Benign CNVs have been proposed as major factors responsible for human diversity. Moreover, it has been recognized that CNVs can even affect the transcriptional activity and translational levels of adjacent genes [6]. It is therefore possible for CNVs that were initially considered benign to later be proven to increase susceptibility to multifactorial diseases or cause genetic diseases with late onset or incomplete penetrance. Clinical variability could also be explained in part by other genetic or environmental determinants, modifying factors of other genes, multigenic inheritance, imprinting, and unmasking of recessive genes. In 2015, Zarrei et al. compiled a CNV map of the human genome and estimated that 4.8–9.5% of the human genome consists of CNVs; they further identified approximately 100 genes whose loss is not associated with any severe consequences [7]. Previously, most CNV-focused population studies have been conducted on clinically enriched populations with various conditions, such as cancer, obesity, idiopathic male infertility, Alzheimer’s disease, schizophrenia, epilepsy, intellectual disability, autism, and even prion diseases [1,8,9,10]. To our knowledge, genome-wide analyses of CNVs in large, healthy populations are still insufficient or lacking, unlike genomic variation studies focused on single-nucleotide variations (SNVs). The vast majority of CNV data is derived from individuals of European descent residing in Western countries, which may account for the underestimation of genomic variants in other populations [11].

Although there are certainly many relatively common CNVs, herein we present a detailed analysis of the 20 most frequently observed maternal CNVs larger than 200 kb (average size 431 kb, median size 340 kb) in a cohort of pregnant women analyzed via genome-wide NIPT. These data were compared with publicly available databases, including gnomAD SVs v2.1, dbVar, and DGB, especially concerning European non-Finnish populations. A subset of findings of unknown or conflicting significance was assessed, with an emphasis on genomic content. Our overview has been focused on a Central/Eastern European population (Slovakia, Czech Republic, and Hungary).

## 2. Materials and Methods

### 2.1. Patients and Sample Collection

Pregnant women who underwent NIPT as either first-tier or second-tier screening starting from the 11th gestational week were considered for this study. All participating women were recruited from prenatal obstetric centers across Slovakia, the Czech Republic, and Hungary between 2016 and 2019. Twin pregnancies were not excluded. All women provided signed informed consent for inclusion in the study before participation. It was anticipated that all participants would be clinically healthy or at least without known genetic abnormalities at the time of pregnancy. The study was conducted in accordance with the Declaration of Helsinki, and the protocol was approved by the Ethics Committee of the Bratislava Self-Governing Region on 30 June 2015 (03899/2015/HF). The results, unless truly pathogenic, were not disclosed to the participants.

Ten milliliters of maternal peripheral blood was collected into a blood tube containing EDTA or a Cell-Free DNA BCT tube (STRECK, La Vista, NE, USA). Plasma was prepared within 36 h after collection (a longer time was acceptable for STRECK) using a two-step centrifugation protocol. The whole blood sample was first centrifuged at 1600× *g* for 10 min at 4 °C, followed by a subsequent centrifugation step at 16,000× *g* for 10 min. All subsequent molecular tests, including cell-free DNA isolation, modified genomic library preparation with Illumina TruSeq Nano chemistry, and DNA sequencing, were performed as previously described [12].

### 2.2. Bioinformatic Analysis for CNVs

Sequencing reads were aligned to the hg19 reference (NCBI build 37) using the Bowtie2 algorithm [13]. Read counts were collected per 20 kb bin. Then, two-step normalization was applied, which included locally weighted scatterplot smoothing (LOESS) [14] and PCA normalization to remove higher-order population artifacts on autosomes [15]. Finally, the signal was split into regions with equal-level signals using the circular binary segmentation algorithm from the R package DNA copy [16]. The resulting data were visualized using an in-house CNV caller tool [17] (Figure 1). These figures were automatically generated for each chromosome, including X and Y.

The average sequencing depth of our NIPT method for each sample was between 0.12 and 0.5×. A minimal number of sequencing reads of no less than 5 million with no upper limit per sample was obtained using a middle-throughput NextSeq 500/550 sequencer (Illumina, CA, USA), primarily for the analysis of fetal aneuploidy. Segments longer than 200 kbp with abnormal gain or loss with signal deviation exceeding 75% were designated as maternal and annotated using DECIPHER [18] and the X-CNV tool [19]. Each CNV call of genome assembly GRCh37 (hg19) was lifted over to hg38 using a web-based tool [20] featured in the UCSC Genome Browser [21]. Hg38 coordinates were recorded to DECIPHER.

### 2.3. Statistical Analysis

The presence of MOVs in our dataset was compared to population frequencies in the merged databases dbVar/DGV [19] and gnomAD SVs v2.1 [22] for non-Finnish European populations and then evaluated using chi-squared statistical tests. DGV and gnomAD variants represent a curated set of variants from selected studies with high resolution and quality evaluated for accuracy and sensitivity. Therefore, an overlap with DGV and gnomAD variants indicated that our CNV calls were likely to be true positives. For two particular variants, only partial overlaps were found in gnomAD, and a test was performed for cases when these were counted as matches.

The number of observed variants per megabase was calculated for each chromosome. The effective lengths of chromosomes (excluding unmappable regions) were used for this calculation. Normal distribution of this value across chromosomes was assumed, and the mean and standard deviation were estimated. The probability of the value for each chromosome was assessed with respect to the distribution, and potential outliers (chromosomes with a significantly different ratio of variants per megabase) were identified.

## 3. Results

We evaluated 6422 NIPT analyses, with the number of reads ranging from 5 M (million) to more than 20 M. We identified 942 distinct maternal CNVs based on a detection resolution of 200 kb, of which 328 were detected repeatedly in at least two samples (Appendix A). This means that approximately one in seven samples contained at least one maternal CNV larger than 200 kb. There were 659 duplications and 283 deletions. Any CNVs that occurred two or more times were referred to as multiple occurring variants (MOVs). We identified 328 distinct MOVs, including 92 deletions and 236 duplications (Appendix A). Some samples carried as many as 2–3 MOVs. The chromosome with the highest number of variants was chr2 (71), and the chromosome with the lowest number of variants was chr19 (8). In our dataset, we aimed at the 30 most frequent MOVs, which occurred from 12 times (del 2p22.3, dup 4q35.2, del 4q35.2, del 6q26, del 7q31.1, and Xp21.1) to 126 times (dup 6q27) (Table 1, Figure 2).

The frequency of the most common MOV ranged from 1.96% (dup 6q27) to 0.19% (del 2p22.3; dup/del 4q35.2; del 6q26; del 7q31.1; Xp21.1). The first two MOVs—dup 6q27 (Figure 2) and dup 22q11.22—were CNV polymorphisms, as their frequency was higher than 1%: 1.96% and 1.53%, respectively. In gnomAD SVs v2.1 (European), we found no match in eight cases; the same applied to dbVar/DGV, with overlap for dup 12q24.13–q24.21, dup 3p26.3, dup 4q35.2, and del 6q26.18. These MOVs contain protein-coding genes, some of which, such as *NIPA1* (OMIM 608145) and *PRKN* (OMIM 602544), are pathogenic. Ten intergenic duplications or deletions were without gene content; however, they did contain regulatory elements, such as enhancers, promoters, transcription factors (TF), binding sites, etc. After applying ACMG guidelines [23], almost all said variants were variants of uncertain significance (VUS) that have no known clinical relevance. Dup 15q13.3, dup/del 15q11.2, del 6q26, and del 7q31.1 have rather conflicting interpretations due to reduced penetrance and variable expressivity. According to artificial intelligence integrated in the X-CNV predictive tool [19], we identified 37 MOVs out of the total of 328 (11%) classified as likely pathogenic/pathogenic (Table 2) [24,25,26,27].

We assessed genome variability according to the size of each chromosome, but size did not seem to be significant. On the other hand, there seemed to be a correlation between chromosomal size and the number of variants, except for chromosome 19, which had fewer variants relative to its size (Table 3).

## 4. Discussion

We identified the frequencies of CNVs detected in the selected Central/Eastern European countries and compared them to the CNV frequencies of non-Finnish European populations in the genomic variants databases (DGV and dbVar) and the gnomAD SVs v2.1 database. The sizes of the respective cohorts were comparable for the purposes of this paper: 11,222, 7624, and 6422 for DGV, gnomAD, and our laboratory in-house database, respectively. When choosing population databases, it is necessary to consider that different methods for CNV detection could lead to varying sizes of identical CNVs [28]. DGV was created in 2004 as a comprehensive catalog of human-contained data of array comparative genomic hybridization (aCGH), with the gradual addition of sequencing data, while gnomAD SVs v2.1 was introduced later and is solely based on high-throughput sequencing. We applied merged population databases, with a virtual unification of CNVs in DGV and dbVar coming from different genomic platforms, as described in detail in Zhang et al. (2021).

The MOV that recurred most often in our dataset was the 6q27 duplication (1.96%), whose frequency was not significantly different from that found among non-Finnish Europeans in the dbVar/DGB database (3.2%) and gnomAD SVs v2.1 (1.4%). Thus far, the three coding genes *AFDN*, *FRMD1*, and *KIF25* in this locus are not known to be associated with any particular condition. Nevertheless, the genomic region contains at least 141 regulatory elements (enhancers, promoters, TF binding sites, etc.); hence, the likely gain is of unclear phenotypical effect (VUS) rather than benign consequences.

The 22q11.22 region overlaps two protein-coding genes, *PPM1F* (OMIM 619309) and *TOP3B* (OMIM 603582), and more than 30 other genes, mainly from the immunoglobulin lambda variable (IGLV) family of functional genes, pseudogenes, and vestigial sequences interspersed in the IGLV locus [29]. We hypothesized that this polymorphic MOV might be responsible for specific antibody diversity in our region. However, further functional investigations are necessary to confirm this hypothesis. We did not register any overlap with gnomAD SVs v2.1, and we anticipated sequence artifacts, but after all, it is a relatively common MOV in the non-Finnish European population in the DGV database (0.7%).

The region of chromosome 15q11–13 is susceptible to genomic rearrangements. Its genomic instability has been attributed to a high density of low-copy number repeats (LCR) mediating aberrant interchromosomal exchanges during meiosis by non-allelic homologous recombination (NAHR) [30].

Small duplications of 15q15.3 containing the *CHRNA7* and *OTUD7A* genes have been reported to yield highly variable phenotypes, ranging from normal to various neurological manifestations. Microduplications of 15q13.3 are equally prevalent in clinical cases and in the general population, making it difficult to assess their contribution to pathogenicity [31]. Speech delay, autistic behaviors, and muscle hypotonia occur in all affected patients, whereas intellectual disability, developmental delay, and epileptic seizures are less common (60%). The cholinergic nicotinic receptor subunit alpha 7 (*CHRNA7*) gene is a clear candidate for behavioral disorders [24]. Our observations indicate that there is a significant difference between our prevalence (0.6%) and the prevalence of the European gnomAD (0.23%); however, the difference between our prevalence and that found in dbVar/DGV (0.8%) is less notable.

Identically frequent duplication and threefold larger reciprocal deletion of the 4q35.2 region encompassing the three genes *ZFP42* (OMIM 614572), *TRIML2*, and *TRIML1* (none of which is morbid) are not associated with any known clinical condition. Surprisingly, they were relatively common in our cohort, while there were no matches in the dbVar and DGV population databases. Moreover, the phenomenon is extremely rare in all gnomAD SVs subpopulations. Applying the ACMG guidelines and the X-CNV predictor, this MOV has been determined to be a variant of unknown significance; however, it has an abundance of regulatory elements. We speculate that both are candidates for population-specific variants; unfortunately, there is not yet any further knowledge regarding medical consequences or perhaps being an adaptive trait.

We recorded no matches in the above-mentioned databases concerning dup 12q24.13–q24.21 (hg19: chr12:114,260,000–14,540,000) and dup 3p26.3 (hg19: chr3:2,660,000–3,020,000). Additionally, 3p26.3 contains the dosage-sensitive gene *CNTN4* (OMIM 607280), recently recognized as a risk factor for autism spectrum disorder and other neuropsychiatric disorders of unknown etiology [32,33]. Larger cohorts are needed for comprehensive and unbiased phenotyping and molecular characterization that may lead to a better understanding of the underlying mechanisms of reduced penetrance, variable expressivity, and potential parent-of-origin effect of copy number variations encompassing *CNTN4*.

Duplication of the 19q13.41 locus contains three OMIM genes. *ZNF350* is an important gene in human mammary oncogenesis that interacts with *BRCA1*. Missense polymorphisms can influence the transcriptional activity of the tumor suppressor gene *BRCA1*, increasing a woman’s risk of developing breast cancer [34]. On the other hand, other studies have shown that overexpression of *ZNF350* significantly impaired the migration of tumor cells in colorectal cancer and inhibited growth and metastatic activity in cervical cancer, acting as a potent tumor suppressor in different types of cancer [35,36]. *FPR1* and *FPR2* are G-coupled receptors expressed in various immune cells that are involved in many pathological processes. Several genes from the zinc finger family, including *ZNF577*, *ZNF649*, *ZNF615*, *ZNF614*, and *ZNF432*, are under intense investigation due to their altered methylation status in various cancer types [37,38,39].

According to genomic content, dup/del 15q11.2 in our dataset is potentially clinically relevant; however, interpretation is rather conflicting due to low penetrance and reduced expressivity. 15q11.2 is a region prone to chromosomal rearrangement in terms of gains and losses. An Israeli collective led by Maya et al. assessed the overall prevalence vs. penetrance in prenatal and postnatal clinically indicated cases and observed no significant difference between the indicated and healthy populations. Their frequencies compared to our observations are not significantly different for duplication (0.78% vs. 0.54%, respectively), unlike deletions, which were less frequent in our pregnant cohort (0.49% vs. 0.2%). Of course, our observation could be biased with respect to different methodologies used as well as the population’s origin. Duplications occurred almost twice as often as deletions. However, the phenotypical penetrance was found to be only 1.16% for duplications and 2.18% for deletions [25]. The less frequent microdeletion 15q11.2, containing four protein-coding genes (*NIPA1*, *NIPA2*, *CYFIP1*, and *TUBGCP5*), is associated with Burnside–Butler syndrome (BBS), which partially overlaps with certain neurodevelopmental disorders, including Prader-Willi and Angelman syndrome [40]. Twice as common is the “inverted” BBS region, reciprocal to the deletion, which seems to be relatively common (0.54% vs. 0.3%) in DGV but has no match in gnomAD. Interestingly, we found no matches in gnomAD SVs v2.1 for both deletion and duplication of this region. However, the finding is well recognized and relatively frequent. We would expect to find these CNVs across all population databases because they occur even in healthy populations without any phenotype.

Considering that the frequency among carriers was almost 0.5% in this study, the Xp22.31 duplication involving four genes (*PUDP*, *STS*, *VCX*, and *PNPLA4*) appears to be quite common. Reciprocal deletion spanning the *STS* gene is associated with congenital X-linked ichthyosis (OMIM 308100), together with corneal opacities, cryptorchidism, cardiac arrhythmias, and higher rates of developmental and mood disorders, affecting mainly males [41,42,43]. The phenotypic spectrum of male and female carriers can differ significantly, probably due to X inactivation in females. While the phenotypes associated with deletions are reasonably well characterized, phenotypes with reciprocal duplications exhibit a wide range of medical and neurobehavioral disorders, including autism and cognitive impairment, as well as speech and language difficulties. Paradoxically, there is evidence that high levels of the steroid sulphatase have a minor protective effect against depressive/anxiety behavior via higher levels of steroid hormones in carrier individuals compared to unaffected controls. Female carriers are more likely to suffer from gastro-esophageal reflux disease (without esophagitis) than sex-matched controls [44]. We cannot state with certainty that women in our region tend to suffer less from depressive behavior compared to women elsewhere due to the higher prevalence of dup Xp22.31, but it is nevertheless an interesting consideration.

6q26 contains a putative haploinsufficient gene *PRKN* (parkin RBR E3 ubiquitin protein ligase). Deletions involving the *PRKN* gene are associated with Parkinson’s disease, autosomal recessive juvenile Parkinson’s disease (OMIM600116), and autism spectrum disorder. The *PRKN* gene is located at the common *FRA6E* fragile site, and copy number variants, as well as single nucleotide variants, are frequently detected [45,46]. Since the vast majority of cases are recessive, there is still the possibility of unmasking a heterozygous mutation on the second allele.

The *IMMP2L* gene (inner mitochondrial membrane peptidase subunit 2) is implicated in Gilles de la Tourette syndrome (GTS, OMIM 137580), Alzheimer’s disease, autism spectrum disorder (ASD), schizophrenia, and other neurodevelopmental disorders. GTS, a neuropsychiatric disorder manifested by repetitive involuntary motor and vocal tics that fluctuate in severity, occurs in 0.4% to 3.8% of the population worldwide [47]. GTS is thought to be inherited in an autosomal dominant manner with variable expression and reduced penetrance. Gross deletions at the exon level represent the most common type of mutation (~80%). The genotypic background of these neuropsychiatric disorders is highly heterogeneous; expression and the resulting phenotype may be influenced by environmental or even epigenetic factors through different methylation patterns [48]. *IMMP2L* might also regulate processes in the reproductive system. This hypothesis was confirmed by the identification of deletions of the *IMMP2L* gene in patients with primary infertility [49]. Unfortunately, regarding our cohort of pregnant women, we do not have information regarding their inability to conceive naturally, as this information is not mandatory. Nevertheless, it would be valuable to investigate whether women with *IMMP2L* gene deletions are overrepresented in the IVF population.

Thirty-seven MOVs (11%) were likely pathogenic/pathogenic according to X-CNV, whereas only four (1.2%) were classified identically according to ACMG automated guidelines. The rule-based ACMG guidelines are often considered the gold standard in variant classification. Unfortunately, they lack important information regarding variant segregation in families and specific phenotypes. Moreover, ACMG is also prone to subjectivity of the scoring person; therefore, we preferred artificial intelligence (AI) in the computational X-CNV tool to predict pathogenicity, integrating four categories of features: universal, coding region, noncoding, and genome-wide.

We were curious about the known CNVs in which expression follows the parent-of-origin pattern. Genomic imprinting is a classic epigenetic phenomenon that involves transcriptional silencing of one parental gene allele. There is only one such region of interest in our dataset: the 15q11.2 deletion involving the imprinted *NIPA1*, *NIPA2*, *CYFIP1*, and *TUBGCP5* genes, which exhibit neurodevelopmental phenotypes, such as epilepsy, macrocephaly, and autism spectrum disorder, when inherited from the mother, while paternal deletions have been associated with congenital heart disease and abnormal muscle phenotypes [50].

The distribution of all structural variants and MOVs above 200 kb according to chromosomal size was quite even, except for chromosome 19, which seems to have lower variability compared to the other chromosomes. Paradoxically, chr19 has the highest gene density of all chromosomes and the highest GC content, which both predispose it to higher nucleotide variability within and between species [51,52]. However, given the higher gene density, any CNV interspersing this region would logically have a potentially harmful effect.

## 5. Conclusions

Building a CNV variant database using NIPT is a very accessible and convenient way of obtaining genomic data for genome-wide population analyses. Some variants may correlate with susceptibility to certain diseases or represent an evolutionary advantage in the context of adaptation to the environment.

Our dataset has the ambition of enriching the existing and any future population databases, which makes it a valuable building block for further research on the functioning of these variants, whether functional or biological, direct or indirect. We are aware that small maternal imbalances below 200 kb were not included in this study because they are beyond our resolution capabilities. We utilized NIPT analyses of pregnant women generated by low-coverage sequencing as a source of population data on recurrent CNVs (MOVs) without any additional costs. Compared to other frequently used population databases, recurrence often differs significantly, even when compared to non-Finnish European populations. Therefore, some MOVs seem to be candidates for population-specific CNVs associated with the Central/Eastern European region. Based on the mappable chromosomal size, we did not record any significant differences in the occurrence of MOVs, except for chromosome 19. Some MOVs potentially have medical consequences, although they may have low penetrance and expressivity.

## Figures and Tables

**Figure 1 diagnostics-12-03056-f001:**
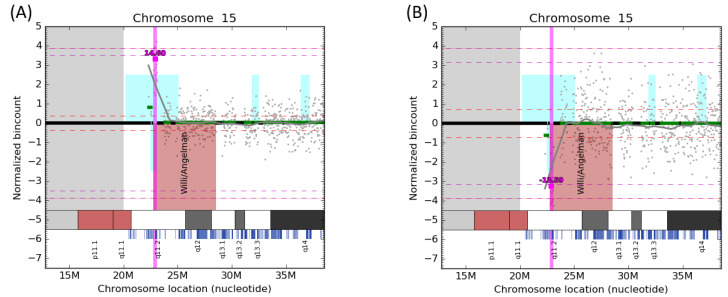
Visualization of CNV in the proximal region of 15q. Normalized read counts per bin are depicted as gray dots. The ***dup 15q11.2*** (**A**)***/del 15q11.2*** (**B**) (approximately 320 kb long) is shown by the vertical line. The light gray vertical band depicts an unmappable centromere region. Black horizontal bands signify bins that did not pass quality metrics (centromere) and were thus excluded from the analysis. The approximated z-score for CNV is displayed over the magenta segment.

**Figure 2 diagnostics-12-03056-f002:**
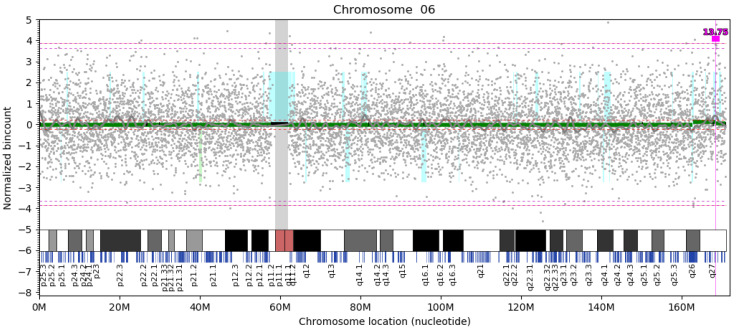
The most frequent MOV dup 6q7 in the dataset of 6422 analyses is localized in the terminus of the long arm of chromosome 6.

**Table 1 diagnostics-12-03056-t001:** The most frequent maternal MOVs in our cohort of NIPT analyses.

CNV	Size	Genomic Coordinates (GRCh37/hg19)	Protein Coding Genes	ACMG Prediction	X CNV Prediction	n	Frequency (%)	dbVAR/ DGV Europe Frequency (%)	gnomAD Europe SV 2v.1 Frequency (%)
dup 6q27	240 kb	chr6:168,340,000–168,580,000	AFDN, FRMD1, KIF25	VUS	VUS	126	1.962	3.1 (NF)	1.4
dup 22q11.22	260 kb	chr22:22,300,000–22,560,000	IGLV11-55, IGLV4-60, IGLV4-69, IGLV6-57, IGLV8-61, PPM1F, TOP3D	VUS	LB	98	1.526	0.7 (NF)	0
dup 8p23.2	200 kb	chr8:2,360,000–2,560,000	-	VUS	B	58	0.903	2.9 (F); 1 (NF)	0.48
dup 11q25	360 kb	chr11:134,360,000–134,720,000	-	B	B	55	0.856	1.9 (F); 0.9 (NF)	0.39
dup 15q13.3	480 kb	chr15:32,020,000–32,500,000	OTUD7A, CHRNA7	VUS	LB	42	0.654	0.8 (NF)	0.23
dup 15q11.2	320 kb	chr15:22,760,000–23,080,000	NIPA1, NIPA2, CYFIP1, TUBGCP5	VUS	VUS	38	0.592	0.3 (NF)	0
dup Xp22.31	1.66 Mb	chrX:6,220,000–8,960,000	PUDP, STS, VCX, PNPLA4	VUS	P	31	0.483	0.1 (NF)	0
dup 1q25.1	320 kb	chr1:175,420,000–175,740,000	TNR	VUS	B	30	0.467	0.1 (NF)	0.06
del 17q22	340 kb	chr17:50,960,000–51,300,000	-	VUS	B	28	0.436	0 (NF)	0.026
del 9p23	260 kb	chr9:11,920,000–12,180,000	-	VUS	B	27	0.420	5.8 (F); 1.2 (NF)	0.01
dup 12p11.1	500 kb	chr12:34,300,000–34,800,000	-	VUS	B	24	0.374	0.1 (NF)	0.026
dup 12q24.13-q24.21	280 kb	chr12:114,260,000–114,540,000	RBM19	VUS	LB	20	0.311	0 (NF)	0
dup 19q13.41	320 kb	chr19:52,280,000–52,600,000	FPR1, FPR3, ZNF577, ZNF649, ZNF613, ZNF350, ZNF615, ZNF614, ZNF432, ZNF841	VUS	LB	20	0.311	0.4 (NF)	0.026
dup 6p11.2	600 kb	chr6:57,400,000–58,000,000	-	VUS	B	20	0.311	0.1 (NF)	0.026
dup 7q11.21	200 kb	chr7:64,680,000–64,880,000	ZNF92	VUS	B	19	0.296	2.9 (F); 0.4 (NF)	0.26
dup2p22.3	680 kb	chr2:32,640,000–33,320,000	BIRC6, TTC27, LTBP1	VUS	VUS	18	0.28	1 (F); 0.1 (NF)	0.18
dup 14q21.2	440 kb	chr14:43,820,000–44,260,000	-	VUS	B	17	0.265	0.4 (NF)	0.15
del 7q11.21	220 kb	chr7:64,680,000–64,880,000	ZNF92	B	B	17	0.265	0.4 (NF)	0.26
dup 3p26.3	360 kb	chr3:2,660,000–3,020,000	CNTN4	VUS	VUS	16	0.249	0 (NF)	0
dup 7q11.21	480 kb	chr7:62,040,000–62,640,000	-	B	B	16	0.249	0 (NF)	0.31
dup Xq27.2	380 kb	chrX: 140,360,000–140,740,000	SPANXA1, SPANXA2	B	B	14	0.218	0.3 (NF)	0.44
del 15q11.2	240 kb	chr15:22,840,000–23,080,000	NIPA1, NIPA2, CYFIP1, TUBGCP5	VUS	P	13	0.202	0.1 (NF)	0.18
del 2p22.3	260 kb	chr2:35,760,000–36,080,000	-	B	B	12	0.187	0.2 (NF)	0.066
del 4q35.2	1.6 Mb	chr4:188,280,000–189,920,000	ZFP42, TRIML2, TRIML1	VUS	VUS	12	0.187	0 (NF)	0.026
dup 4q35.2	480 kb	chr4:188,700,000–189,180,000	ZFP42, TRIML2, TRIML1	VUS	VUS	12	0.187	0 (NF)	0
del 6q26	800 kb	chr6:162,340,000–163,140,000	PRKN	VUS	LP	12	0.187	0 (NF)	0
del 7q31.1	460 kb	chr7:110,840,000–111,300,000	IMMP2L	B	LP	12	0.187	0.2 (NF)	0
dup Xp21.1	240 kb	chrX:33,000,000–33,920,000	-	B	B	12	0.187	0 (NF)	0.017

NF—non Finnish European, F—Finnish European, ACMG—The American College of Medical Genetics and Genomics classification; B—benign, LB—likely benign, VUS—variant of uncertain significance; LP—likely pathogenic; P—pathogenic; n—number of occurence in our dataset (N=6422).

**Table 2 diagnostics-12-03056-t002:** Likely pathogenic/pathogenic MOVs.

CNV	Size	Genomic Coordinates (GRCh37/hg19)	Protein Coding Genes	ACMG Prediction	X CNV Prediction	n	Frequency (%)	dbVAR/ DGV Europe Frequency (%)	gnomAD Europe SV 2v.1 Frequency (%)
del 15q11.2	240 kb	chr15:22,840,000–23,080,000	NIPA1, NIPA2, CYFIP1, TUBGCP5	VUS	P	13	0.202	0.1 (NF)	0.18
del 6q26	800 kb	chr6:162,340,000–163,140,000	PRKN	VUS	LP	12	0.187	0 (NF)	0
del 7q31.1	460 kb	chr7:110,840,000–111,300,000	IMMP2L	B	LP	12	0.187	0.2 (NF)	0
del 20p12.1	540 kb	chr20: 14,640,000–15,180,000	MACROD2	B	LP	11	0.171	0.1	0
del 22q11.21-q11.22	280 kb	chr22: 22,300,000–22,580,000	PPM1F, TOP3B	VUS	LP	8	0.125	1.9 (F); 0.7 (NF)	0
del Xq12	280 kb	chrX:65,640,000–66,020,000	EDA2R	B	LP	8	0.125	0.1	0.017
del 5q23.1	260 kb	chr5:118,860,000–119,120,000	HSD17B4, FAM170A	VUS	LP	6	0.093	0	0.026(partial overlay)
del 5q23.1-q23.2	1.38 Mb	chr5:119,960,000–121,340,000	SRFBP1, PRR16,FTMT	VUS	LB	5	0.078	0	0
dup 5p15.33	1.48 Mb	chr5:40,000–1,520,000	SLC6A3, LRRC14B, ZDHHC11B, ZDHHC11, CCDC127, SLC9A3, TPPP, PLEKHG4B, SLC12A7, SLC6A19, SLC6A18, LPCAT1, PDCD6, TERT, AC026740.1, NKD2, AHRR, CLPTM1L, BRD9, TRIP13, EXOC3, CEP72, SDHA	VUS	P	5	0.078	0	0
del 10q21.3	380 kb	chr10:68,260,000–68,640,000	CTNNA3	VUS	LP	5	0.078	0.1	0
del 4q22.3	320 kb	chr4:98,520,000–98,840,000	STPG2	B	LP	4	0.062	0	0.013 (overlay)
dup 16p13.11	1.16 Mb	chr16:15,120,000–16,280,000	MYH11, NDE1, MARF1, PDXDC1, MPV17L, NTAN1, RRN3, FOPNL, AC140504.1, C16orf45, NPIPA5, ABCC6, ABCC1	VUS	LP	4	0.062	0.2	0
del 3p26.2-p26.1	600 kb	chr3: 3,840,000–4,440,000	SUMF1, SETMAR, LRRN1	VUS	P	4	0.062	0	0
del 3q26.31-q26.32	660 kb	chr3:175,080,000–175,740,000	NAALADL2	B	LP	4	0.062	0.1	0.039
del 5q12.1	860 kb	chr5:59,360,000–60,220,000	ERCC8, FKSG52, DEPDC1B, PDE4D, ELOVL7	VUS	P	3	0.047	0	0
del 8p22	1.4 Mb	chr8:13,860,000–15,260,000	SGCZ	VUS	LP	3	0.047	0	0
del 10p12.31	400 kb	chr10:19,420,000–19,820,000	MALRD1	B	LP	3	0.047	0.1	0
del 16p12.2	480 kb	chr16:21,940,000–22,420,000	EEF2K, CDR2, MOSMO, SDR42E2, POLR3E, UQCRC2, PDZD9, VWA3A	VUS	LP	3	0.047	0	0.039
dup 22q11.21	2.17 Mb	chr22:18,900,000–21,440,000	RIMBP3, PI4KA, KLHL22, AC007326.4, GNB1L, TBX1, TRMT2A, TANGO2, FAM230A, RTL10, GP1BB, AC002472.1, ZNF74, P2RX6, DGCR8, ESS2, CRKL, SLC7A4, TMEM191B, DGCR2, USP41, DGCR6, C22orf39, RTN4R, DGCR6L, MED15, UFD1, TXNRD2, CLDN5, GGTLC3, TSSK2, GSC2, ARVCF, SLC25A1, COMT, CLTCL1, SERPIND1, LRRC74B, AC007731.5, SCARF2, HIRA, CCDC188, RANBP1, THAP7, SNAP29, PRODH, MRPL40, ZDHHC8, CDC45, AIFM3, SEPT5, LZTR1, SEPT5-GP1BB	P	P	3	0.047	0.1	0
dup 5p13.2-p13.1	1.48 Mb	chr5:37,100,000–38,580,000	EGFLAM, CPLANE1, LIFR, NUP155, WDR70, GDNF	LP	P	2	0.031	0	0
del 7p21.2	360 kb	chr7:16,120,000–16,480,000	ISPD	VUS	LP	2	0.031	0	0
del 7q21.11	740 kb	chr7:84,340,000–85,080,000	SEMA3D	VUS	LP	2	0.031	0	0
del 8q22.2	400 kb	chr8:100,340,000–100,740,000	VPS13B	VUS	LP	2	0.031	0.1	0
del 8q24.23-q24.3	1.74 Mb	chr8: 139,540,000–141,280,000	TRAPPC9, COL22A1, KCNK9	VUS	P	2	0.031	0	0
del 9q32	520 kb	chr9:119,220,000–119,740,000	ASTN2, TRIM32	VUS	P	2	0.031	0	0
dup 10q11.22-q11.23	3.9 Mb	chr10:47,640,000–51,580,000	PARG, AL591684.1, TMEM273, ERCC6, AL603965.1, CHAT, VSTM4, NCOA4, GDF2, TIMM23B, ARHGAP22, C10orf71, FAM21B, FAM25C, MAPK8, OGDHL, FAM25G, GDF10, ANXA8, DRGX, ANXA8L2, RBP3, FRMPD2, AGAP8, AGAP9, ZNF488, MSMB, FAM170B, LRRC18, FAM21D, ASAH2C, WDFY4, PTPN20B, SLC18A3, ANTXRL, C10orf53	VUS	P	2	0.031	0	0
del 11p15.3	260 kb	chr11: 11,360,000–11,620,000	GALNT18, CSNK2A3	B	LP	2	0.031	0	0
del 12p13.31	320 kb	chr12:5,860,000–6,180,000	VWF, ANO2	B	LP	2	0.031	0	0
del 12p11.23	480 kb	chr12:27,280,000–27,760,000	SMCO2, ARNTL2, STK38L, PPFIBP1	B	LP	2	0.031	0	0.026
del 18p11.32	800 kb	chr18:1,820,000–2,620,000	NDC80, METTL4	B	LP	2	0.031	0	0
del 20p12.3	480 kb	chr20:8,100,000–8,560,000	PLCB1	VUS	LP	2	0.031	0.1	0
del 1q21.1	880 kb	chr1:144,880,000–145,720,000	ANKRD35, ITGA10, POLR3C, AC239799.1, TXNIP, RBM8A, AC243547.3, PIAS3, RNF115, NOTCH2NLA, ANKRD34A, AL590452.1, POLR3GL, LIX1L, PEX11B, NUDT17, NBPF10, NUDT4B, SEC22B, HJV, CD160, PDE4DIP	VUS	LP	2	0.031	0	0
del 2p16.3	320 kb	chr2: 50,700,000–5,1020,000	NRXN1	LP	P	2	0.031	0	0
del 2p16.3	400 kb	chr2: 51,080,000–51,480,000	NRXN1	LP	P	2	0.031	0	0
del 2p13.2	440 kb	chr2: 72,500,000–72,940,000	EXOC6B	VUS	LP	2	0.031	0	0
del 2q12.3	420 kb	chr2: 108,600,000–109,020,000	SLC5A7, SULT1C3, SULT1C2, SULT1C4	B	LP	2	0.031	0	0
del 3q26.1	800 kb	chr3:164,920,000–165,720,000	BCHE	VUS	LP	2	0.031	0	0

**Table 3 diagnostics-12-03056-t003:** Number of variants across chromosomes.

Chromosome	Effective Length (GRCh38/hg20) *	Number of Variants	Number of Variants per Megabase	*p*-Value (<0.05) *
1	231,223,641	64	0.276788306	0.309486126
2	240,863,511	71	0.294772752	0.39856174
3	198,255,541	67	0.337947679	0.625386241
4	189,962,376	58	0.305323618	0.453773416
5	181,358,067	55	0.303267458	0.442908357
6	170,078,524	60	0.352778226	0.697688611
7	158,970,135	60	0.377429383	0.801517446
8	144,768,136	54	0.37301026	0.78465846
9	122,084,564	41	0.335832792	0.614630158
10	133,263,006	45	0.33767811	0.624020457
11	134,634,058	37	0.274819021	0.300273823
12	133,137,821	34	0.255374466	0.216709904
13	97,983,128	29	0.295969322	0.404743508
14	91,660,769	31	0.338203578	0.626681306
15	85,089,576	26	0.305560343	0.455026684
16	83,378,703	32	0.38379105	0.824340228
17	83,481,871	24	0.287487567	0.36152445
18	80,089,650	24	0.299664189	0.4239761
**19**	**58,440,758**	**8**	**0.136890764**	**0.008989368**
20	63,944,268	15	0.234579275	0.144314325
21	40,088,623	9	0.224502598	0.115899249
22	40,181,019	14	0.348423219	0.677094783
X	154,893,034	84	0.542309734	0.998853971

* Effective length represents the actual chromosome size after removal of unmappable and repetitive genome regions used for mapping (e.g., centromeres and 13p, 14p, 15p, 21p, 22p arms of acrocentric chromosomes); *p*-value < 0.05 is statistically significant.

## Data Availability

Not Applicable.

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
