# Peer review of "Maternal Copy Number Imbalances in Non-Invasive Prenatal Testing: Do They Matter?"

_diagnostics, 2022, doi:10.3390/diagnostics12123056_

Round 1

Reviewer 1 Report

The Authors analyzed low coverage NGS data with the aim to enrich either current general population CNV databases and to search for Central-Eastern Europe possibly specific recurrent CNV variants, by a smart approach based on NGS results from over 6000 NIPT blood samples obtained by healthy  pregnant women.

General comments:

The  Authors should provide some text shortening in both ‘Introduction’ and ‘Discussion’ sections.

Materials and Methods section is well reported. Some details about NGS platforms and commercial NIPT kits adopted could improve comprehension of the applied strategies.

In Discussion section, non-informative CNV data findings could be  reported cumulatively.

The Author could provide a comment about imprinted chromosome regions in which CNVs were found.

Specific comments:

a) page 3, line 28: did women undertaking first-tier or second-tier screening   give comparable results?

b) page 4, line 24: a reference for in-house CNVcaller tool could be  of interest

c) page 5, Figure 1: the two images should be named

d) page 7, line 17: ‘…reduced penetrance and variable expressivity…’ a reference should be added

e) page 9: some details about data reported in Table 3 could be appreciated by non-geneticists readers

f) page 12, lines 1-3: the sentence appears unclear

g) Reference list is duplicated from page 19 to page 24

Author Response

At the outset, I would like to thank you for your helpful review

The Authors analyzed low coverage NGS data with the aim to enrich either current general population CNV databases and to search for Central-Eastern Europe possibly specific recurrent CNV variants, by a smart approach based on NGS results from over 6000 NIPT blood samples obtained by healthy pregnant women.

General comments:

The Authors should provide some text shortening in both ‘Introduction’ and ‘Discussion’ sections.

We shortened the introduction slightly

Materials and Methods section is well reported. Some details about NGS platforms and commercial NIPT kits adopted could improve comprehension of the applied strategies.

We have added some details

In Discussion section, non-informative CNV data findings could be reported cumulatively.

My discussion is based on well-known information about the most common MOVs. I have conducted an in-depth analysis of data from the literature to provide more detailed information on most of them. I understand that it can be a bit exhausting to go through all of these VUSs in detail, however, the main focus of my paper was just to focus on the most common MOVs. I can only hope that this is acceptable to you.

The Author could provide a comment about imprinted chromosome regions in which CNVs were found.

This regards only 15q11.2 from all detected MOVs, I have added specific information based on parent-of-origin resulting phenotype

Specific comments:

  1. a) page 3, line 28: did women undertaking first-tier or second-tier screening   give comparable results?

Unfortunately this specific information is not always available individually, women often undergo NIPT before completing biochemical screening and we have no way to obtain this information retrospectively. Situation is also different even in Central Europe countries e.g. in Czech republic first tier screening (FTS) is preferred, in Slovakia second tier screening (STS) prevails. However, we assume that this does not affect the incidence of maternal imbalance. Nevertheless, in the context of fetal findings (T21, T18...) it might be interesting to compare the sensitivity or false positivity of FTS and STS towards detection of fetal aneuploidies.

  1. b) page 4, line 24: a reference for in-house CNVcaller tool could be of interest

reference added

  1. c) page 5, Figure 1: the two images should be named

added

  1. d) page 7, line 17: ‘…reduced penetrance and variable expressivity…’ a reference should be added

reference added

  1. e) page 9: some details about data reported in Table 3 could be appreciated by non-geneticists readers

an explanatory note has been added below the table

  1. f) page 12, lines 1-3: the sentence appears unclear

information was not so important for the context, I have completely removed it

  1. g) Reference list is duplicated from page 19 to page 24

reference list was updated and corrected

Reviewer 2 Report

The study is well documented; the number of patients is large according to the author’s intention to develop a population database in the region of interest for further research in genes an chromosomes variants, whether functional or biological.

The aim of the study is well defined, the design is proper and the analysis of the data is clear and exhaustive.

There is no plagiarism detected, the bibliography is properly chosen.

I could not detect any self citations.

The references are not numbered in the text and in the references, according to the Authors instructions: “References: References must be numbered in order of appearance in the text (including table captions and figure legends) and listed individually at the end of the manuscript. We recommend preparing the references with a bibliography software package, such as EndNoteReferenceManager or Zotero to avoid typing mistakes and duplicated references. We encourage citations to data, computer code and other citable research material. If available online, you may use reference style 9. Below”.

Author Response

Thank you for your review, I followed your reference guidance. By default, we use the Mendeley citation tool, however the numbering in order of occurrence was available
